# Genetics, Treatment, and New Technologies of Hormone Receptor-Positive Breast Cancer

**DOI:** 10.3390/cancers15041303

**Published:** 2023-02-18

**Authors:** William Sebastian, Lauren Forchette, Kelsey Donoughe, Yibei Lun, Anisha Verma, Tuoen Liu

**Affiliations:** West Virginia School of Osteopathic Medicine, Lewisburg, WV 24901, USA

**Keywords:** breast cancer, hormone receptor, estrogen receptor, endocrine therapy, drug treatment, genetics

## Abstract

**Simple Summary:**

In this review paper, we focused on the discussion of various important aspects of hormone receptor (HR)-positive breast cancer, including HR structure and signaling, genetics (epigenetics and gene mutations), gene expression-based assays, traditional and new drugs for treatment, and new technological uses in diagnosis and treatment. Particularly, we summarized the commonly mutated genes and abnormally methylated genes in HR-positive breast cancer and compared the common gene expression-based assays that are used in breast cancer as prognostic and/or predictive tools in detail. All of these topic discussions have not been fully described and summarized within other research or review articles.

**Abstract:**

The current molecular classification divides breast cancer into four major subtypes, including luminal A, luminal B, HER2-positive, and basal-like, based on receptor gene expression profiling. Luminal A and luminal B are hormone receptor (HR, estrogen, and/or progesterone receptor)-positive and are the most common subtypes, accounting for around 50–60% and 15–20% of the total breast cancer cases, respectively. The drug treatment for HR-positive breast cancer includes endocrine therapy, HER2-targeted therapy (depending on the HER2 status), and chemotherapy (depending on the risk of recurrence). In this review, in addition to classification, we focused on discussing the important aspects of HR-positive breast cancer, including HR structure and signaling, genetics, including epigenetics and gene mutations, gene expression-based assays, the traditional and new drugs for treatment, and novel or new uses of technology in diagnosis and treatment. Particularly, we have summarized the commonly mutated genes and abnormally methylated genes in HR-positive breast cancer and compared four common gene expression-based assays that are used in breast cancer as prognostic and/or predictive tools in detail, including their clinical use, the factors being evaluated, patient demographics, and the scoring systems. All these topic discussions have not been fully described and summarized within other research or review articles.

## 1. Breast Cancer Classification and Hormone Receptors

### 1.1. Breast Cancer Classification 

Cancer is a major public health problem, being the second leading cause of death in the United States (US) and worldwide. Breast cancer is the most frequent malignancy and the second leading cause of cancer-related deaths in women worldwide, with ~2.26 million new cases and 685,000 deaths globally in 2020 [1]. The exact etiology of breast cancer is unclear, but some of the risk factors include increased age (most important), family history, early menarche, late menopause, older age at first live childbirth, prolonged hormone replacement therapy, previous therapeutic chest wall irradiation, benign proliferative breast disease, increased mammographic breast density, obesity after menopause, smoking, diabetes, drinking alcohol, nocturnal schedules, and genetic mutations such as BRCA1 and BRAC2 [2,3,4]. The classification of breast cancer can be based on molecular receptor expression, and the receptors include estrogen receptor (ER), progesterone receptor (PR), and human epidermal growth factor receptor 2 (HER2), with the proliferation index marker Ki67. There are four main subtypes of breast cancer (luminal A, luminal B, HER2-positive/enriched, and basal-like/triple-negative) based on receptor gene expression profiling (Table 1) [5,6]. However, the appropriateness of using Ki-67 as a proliferation marker to differentiate subtypes has been questioned. The application of a Ki67 score of positive/negative or high/low in patient follow-up and treatment is controversial, and there is no consensus on this issue today [7]. Understanding the breast cancer classification and subtypes allows for a better and more personalized therapy that targets the exact molecular and pathological mechanisms of the tumor. For example, different therapies, such as endocrine or HER2-targeted therapy, can be used based on the subtypes. In this review, we will focus on the discussion of the hormone receptor-positive (HR-positive) breast cancer, which includes the luminal A and luminal B subtypes.

### 1.2. Hormone Receptor Structure and Function 

Hormone receptors, including estrogen, and progesterone receptors, a subtype of nuclear receptors, play crucial roles in breast cancer. In addition to its major function in the development and maintenance of normal sexual and reproductive functions, estrogen has many other functions, including the regulation of bone density, brain function, cholesterol metabolism, inflammation, and cell proliferation. The term estrogen refers to multiple steroids, including estrone (E1), estradiol (E2), and estriol (E3), and all forms can bind with varying affinities to ERs [8,9,10]. The downstream effects of estrogen are mediated by its binding to either the *ESR1* gene (encoding for ER-alpha or ERα) or the *ESR2* gene (encoding for ER-beta or ERβ). ERα is located on chromosome 6q25.1 and is ~66 kDa, containing 595 amino acids, while ERβ is located on chromosome 14q23–24 and is ~54 kDa, containing 530 amino acids [10]. The comparison of structures of the two ER isoforms is illustrated in Figure 1A. Each ER isoform is composed of five major domains: the amino-terminal domain (NTD, A/B region), DNA-binding domain (DBD, C region), hinge domain (D region), ligand binding domain (LBD), and the carboxyl-terminal domain (CTD, E, and F regions). The D domain/region contains a nuclear localization signal (NLS) and links the DBD and CTD, which contain LBD. Two activation function (AF) domains, AF1 and AF2, located within the NTD and LBD, respectively, are responsible for regulating the transcriptional activity of ER [8,9,10,11]. Both ER subtypes are widely expressed in various tissue types, but there are notable differences in their expression patterns [12]. Specifically, in the human mammary glands, Erα-positive cells are present in ducts and lobules but not in stromal cells. ERβ is present in luminal, myoepithelial, and stromal cells. Studies in ERα knockout mice demonstrated that ERα is required for normal mammary gland growth and maturation. ERβ knockout, however, has little effect on mammary gland development [13]. A variety of studies indicate that ERα is a primary mediator of estrogenic actions in breast cancer [14].

As a steroid derivative, estrogen is lipophilic and can interact with membrane-bound ER or fuse into the cell and interact with cytoplasmic ER. Generally, estrogen signaling regulates gene expression by stimulating ER directly binding to DNA sequences of estrogen response element (ERE) or indirectly by other mechanisms [10]. The general mechanisms of the ER signaling pathway are illustrated in Figure 1B. In the direct signaling pathway, estrogen binding causes conformational changes and the dimerization of ER, which then proceeds to the nucleus to bind to chromatin, commonly at the ERE sequences close to the promoters of ER-targeted genes (Figure 1B). The indirect signaling pathway, also termed “transcriptional cross-talk”, comprises ~35% of estrogen’s gene targets. ER interacts with other response elements and transcription factors, but it does not bind to ERE as with the direct signaling pathway [15]. One example of an indirect pathway is the regulation of stimulating protein-1 (SP-1) via ER activation. Estrogen binding enhances SP-1′s ability to bind onto GC-rich sites on DNA, which stimulates SP-1 to induce genes for progesterone receptor B, LDL, and nitric oxide synthase [8,9,10]. Despite both receptor isoforms working to mediate estrogen functioning via the regulation of transcription factors upon estrogen-ligand binding, their functions are not exactly the same. In general, ERα and ERβ have different functions in carcinogenesis and tumor progression, with ERα acting as an oncogene and ERβ as a tumor suppressor. For example, studies have revealed the antagonistic effects between the two receptor isoforms. In a xenograft model utilizing the HR-positive breast cancer MCF-7 cells, estrogen stimulation increased tumor formation due to the estrogenic effects of ERα. In contrast, in the same model, adding ERβ to the cells prevented tumor formation by estrogen stimulation via inhibiting the expression of transcription factors (e.g., c-myc, cyclin D1, cyclin A) and increasing the expression of P21 and P27 (causing cell cycle arrest), indicating the antagonistic effects of ERβ against ERα [16,17].

Although ERs are more commonly known as nuclear receptors, as described above, the G-protein coupled estrogen receptor (GPER, or GPR30) is an alternate ER that has a distinctly different structure from both ERα and ERβ. A schematic overview of estrogen-GPER interaction and activation is illustrated in Figure 1C [18]. Studies found that GPER is expressed in over 50% of breast cancer cases and is associated with tamoxifen resistance in ER-positive breast cancer. Upon estrogen binding, GPER can lead to downstream effects, such as activating the tyrosine kinase Src and epidermal growth factor receptor (EGFR), which further activates the MAPK and PI3K/AKT pathways [8,19,20]. Studies have shown that GPER expression is strongly associated with the prognosis of ER-positive cancer, e.g., high GPER expression is correlated with a shorter overall patient survival time [21]. EGFR is also associated with causing resistance to endocrine therapy, which can be a potential drug target for drug development [20].

Progesterone, similar to estrogen, is essential for the regulation of normal reproductive functions. In females, progesterone plays a vital role in mammary development/maturation and ovulation. In males, progesterone influences spermiogenesis and testosterone production from Leydig cells. PR has also been shown to play an essential role in triggering ovulation and attenuating ovulatory inflammation [22,23]. The effects of progesterone are mediated through the PR, which has two main isoforms (PR-A ~ 94 kDa and PR-B ~116 kDa), both acting as nuclear receptors similar to ER. [24,25]. PR-A and PR-B isoforms share similarities in structure, which includes the amino-terminal domain (NTD), DNA binding domain (DBD), ligand-binding domain (LBD), and activation functional (AF) domains [26]. The basic structure and activation patterns of PR are illustrated in Figure 1D [24]. Although they display commonality in their structure, the two PR isoforms also have distinct functions: PR-B primarily transactivates progesterone-targeted genes, while PR-A inhibits chromatin binding and ultimately suppresses PR-B function [25].

While the physiological function of progesterone and PR signaling pathways have been thoroughly studied, the precise oncogenic mechanisms of PR in breast cancer are limited because of two main factors. First, it is difficult to distinguish the effects of progesterone and PR from other hormones that influence breast cancer, including growth factors, prolactin, and especially estrogen, as tissues that express PR, such as mammary epithelial tissues, also express ER. Second, only ~1% of breast cancer cases appear to be ER-negative and PR-positive, causing the study of PR to be overlooked. In addition, the number of those cases continually decreases as molecular biology techniques, such as immunohistochemistry and quantitative PCR (qPCR), are being optimized [27,28,29]. 

**Figure 1 cancers-15-01303-f001:**
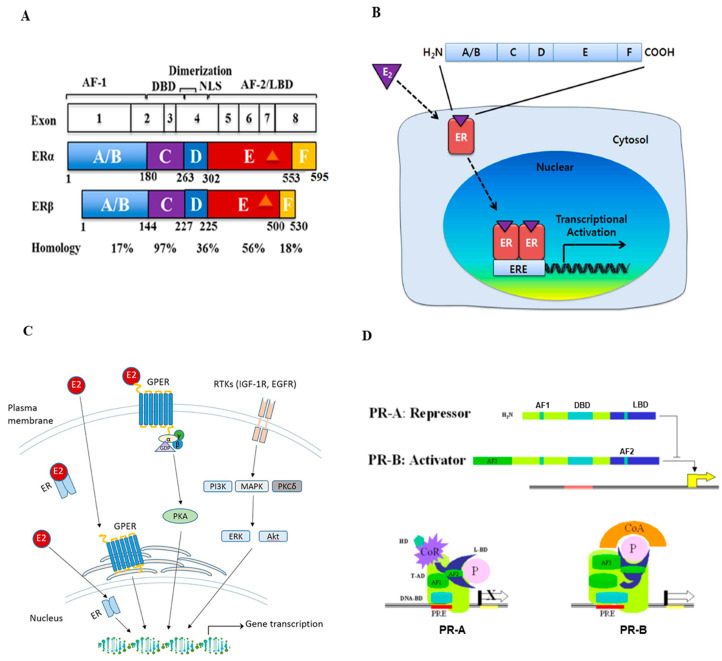
Structure and signal transduction of estrogen receptor (ER) and progesterone receptor (PR). (**A**) Schematic representation of the ER structural regions. Both ERα and ERβ genes are expressed from 8 exons and have five interconnected segments of functional domains. The number of amino acids and percent homology between the two ERs for each segment are indicated (Figure reproduced from [11]). (**B**) Mechanisms of estrogen receptor signaling pathway. Estrogen (E2) first binds ERs, then ERs dimerize and translocate into the nucleus. These complexes bind to estrogen response elements (EREs) and regulate downstream gene transcription (Figure reproduced from [10]). (**C**) Schematic overview of estrogen and G-protein coupled estrogen receptor (GPER) interaction and activation in estrogen signaling pathways. The GPER is an alternate ER with seven-transmembrane domains that mediate nongenomic estrogen-related signaling. Membrane estrogen can interact with GPER, which further activates downstream protein-kinase cascades (Figure reproduced from [18]). (**D**) The structure and activation of PR major isoforms. PR-A is the truncated form lacking the first 164 amino-terminal and is transcriptionally inactive. PR-A also lacks a third transactivation domain (AF3) located in the truncated area, which is known to repress transcriptional activity mediated by PR-B and some other steroids. PR-B binds through its DBD to the progesterone response element (PRE) on the promoter and functions as an activator of the progesterone response gene (Figure reproduced from [24]).

### 1.3. Endocrine Therapy and Mechanisms of Resistance

Endocrine therapy, or hormone therapy, is the basis of drug treatment and the first-line therapy over chemotherapy in most HR-positive breast cancer patients. Currently, the most common endocrine therapy agents used in breast cancer include selective estrogen receptor modulators (SERMs), aromatase inhibitors (AIs), selective estrogen receptor degraders (SERDs), and ovarian function suppression (OFS). However, treatment protocol differs between patients who are postmenopausal and pre- or perimenopausal.

SERMs are commonly used as the first-line agents in ER-positive breast cancer treatment regimens. A meta-analysis found that the absolute risk reduction was ~nine times better in invasive breast cancer patients treated with SERM for 5 years, compared to those who were not treated [30]. SERMs exert their therapeutic effects by modifying the expression of genes whose transcription is regulated by ER signaling. Examples of SERMs commonly used for breast cancer treatment include tamoxifen and toremifene. Tamoxifen has antagonistic effects in breast tissue but has partial agonistic effects in other tissues such as the uterus, heart, and bones. When tamoxifen binds to ERα, it prevents proper LBD interactions, causing the inactivation of the transcriptional region. Additionally, it also recruits corepressors to the ERE of target genes, resulting in ERα silencing [31]. Toremifene has a similar mechanism of action as tamoxifen, acting as an ER antagonist on breast tissue but has agonistic effects on the bone and uterine tissue.

In postmenopausal breast cancer, the first-line option is usually an AI (such as anastrozole, letrozole, or exemestane) which is effective in preventing relapse and prolonging survival. AIs function by inhibiting the enzyme aromatase, which leads to a decrease in the conversion of androgen into estrogen. In a comparative clinical trial, postmenopausal patients with HR-positive breast cancer who received AIs showed a much lower recurrence rate compared to those who received tamoxifen [32]. AIs are normally not used and have not been shown to be efficacious in premenopausal women due to reduced feedback of estrogen to the hypothalamic-pituitary-ovarian (HPO) axis that increases gonadotropin secretion, which stimulates the ovary to produce more estrogen. However, premenopausal women treated with ovarian suppression can benefit from AIs [33,34].

Recently, the effectiveness of SERDs has been examined as a potential bypass in the drug resistance seen in ER-positive breast cancer. Fulvestrant is the first SERD approved by the US FDA in 2002 and works by binding ER and preventing ER translocation to the nucleus for transcriptional regulation. It also degrades mutant ER through the ubiquitin-proteasome system. A clinical trial found that individuals with an ER mutation had a better survival rate with fulvestrant compared to other endocrine therapies [35]. Some SERDs are being investigated in clinical trials; for example, LY3484356 is currently being tested against ER-positive advanced breast cancer and endometrial endometrioid cancer by Eli Lilly and Company [36].

Ovarian function suppression (OFS) is a therapeutic method of limiting estrogen production to minimize the estrogenic effects on ER-positive breast cancer. This is achieved by a temporary suppression of ovarian estrogen synthesis by luteinizing hormone-releasing hormone (LHRH) agonists and permanent interruption of ovarian estrogen synthesis with oophorectomy or radiotherapy. OFS, often preceded by chemotherapy, also allows the use of AIs in premenopause, as mentioned above [37,38].

However, resistance has become an increasing concern with endocrine therapy for breast cancer, which is commonly due to ER mutations causing ER to be constitutively activated even in the absence of ligand interaction [39,40]. Other factors, such as the upregulation of CDK6 and CCND1 genes and the mutation of the PIK3CA gene, are known to cause endocrine therapy resistance to breast cancers [41,42,43]. In addition to endocrine therapy, adjuvant chemotherapy is needed in some HR-positive breast cancer patients. While most ER-positive breast cancers may initially respond to endocrine therapy, ~15–20% of tumors are intrinsically resistant to treatment, and another ~30–40% develop resistance over many years. Resistance to treatment inevitably results in relapse and metastasis, leading to death [44]. The proper time and situation to utilize adjuvant chemotherapy depend largely on patients’ clinical, pathological, and genetic profiles. Some gene expression-based assays are being used to guide the decision to use adjuvant chemotherapy, which will be discussed in Section 3.

## 2. Genetics of HR-Positive Breast Cancer

### 2.1. Gene Mutation in HR-Positive Breast Cancer

All cancers carry somatic mutations in their genomes. Banerji et al. performed whole-genome and whole-exome sequencing of 108 primary, treatment-naïve, breast cancer/normal DNA pairs from all major expression subtypes, including 38 luminal A and 22 luminal B subtypes. They identified six significantly mutated genes in all cases, including PI3KCA, TP53, AKT1, GATA3, MAP3K1, and CBFB [45]. Another group, Ellis et al., reported their whole-genome analysis results of HR-positive breast cancer in response to aromatase inhibitor treatment in the same volume of the *Nature* journal published in 2012. They conducted massively parallel sequencing (MPS) on 77 samples accrued from two neoadjuvant aromatase inhibitor clinical trials. A total of 46 cases underwent whole-genome sequencing, and 31 cases underwent exome sequencing, followed by extensive analysis for somatic alterations and their association with aromatase inhibitor response. From this study, 17 significantly mutated genes were identified in human luminal breast cancer patients, including PIK3CA, TP53, GATA3, CDH1, RB1, MLL3, MAP3K1, CDKN1B, TBX3, RUNX1, LDLRAP1, STNM2, MYH9, AGTR2, STMN2, SF3B1, and CBFB. They also reported that mutant MAP3K1 was associated with luminal A status, low-grade histology, and low proliferation rates, whereas mutant TP53 was associated with the opposite pattern. Mutant GATA3 was correlated with the suppression of proliferation upon aromatase inhibitor treatment [46]. Subsequently, in 2018, the same group led by Griffith and Ellis et al. reported the targeted sequencing results of 83 genes using DNA from primary HR-positive breast cancer patient samples to determine the interactions between somatic mutation and prognosis. They found that mutations of MAP3K1 and PIK3CA were associated with a favorable prognosis and the mutations of TP53, PIK3R1, and DDR1 were associated with a poor prognosis [47]. Whole-exome and transcriptome analyses of patients with ER-positive metastatic breast cancer identified mutations in ESR1, affecting its ligand-binding domain (LBD), suggesting that activating mutations in ESR1 is a key mechanism in acquired endocrine resistance in breast cancer therapy [48,49]. ESR1 mutations express a unique transcriptional profile that favors tumor progression, suggesting that selected ESR1 mutations may influence metastasis. Several research groups have used sensitive detection methods using patient liquid biopsies to track ESR1 or truncal somatic mutations to predict treatment outcomes and tumor progression, and these techniques may be used to guide patient treatment in the future [50]. BRCA1 and BRCA2 (BReast CAncer genes 1 and 2) are well-known tumor suppressor genes linked to breast cancer. Although there is some biological evidence of interactions between estrogens and BRCA proteins, the precise relationship between the two genes and ER remains unclear. Most breast cancers with BRCA1 mutation carriers are triple negative, but only ~10–20% are ER [51]. In contrast, based on Metcalfethe et al. report, the majority (~77%) of breast cancers diagnosed in females with a BRCA2 mutation are ER-positive. BRCA2 tumors are significantly more likely to be ER-positive compared to both BRCA1 and sporadic tumors [52]. The TP53 gene is a well-studied tumor suppressor gene commonly mutated in multiple cancer types. A more recent study by Ji et al. detected 12.8% of ER-positive/HER2-negative breast cancer patients carrying TP53 mutations [53]. Based on the findings of these above studies, the significantly mutated genes identified in HR-positive breast cancer and their functions are summarized in Table 2.

### 2.2. Epigenetic Regulation

In addition to somatic gene mutations, the epigenetic regulation of gene expression also plays an important role in breast cancer development and progression. The term epigenetics refers to the alteration of DNA expression via nonmutational events, which include two common types: DNA methylation and histone modification. DNA methylation has long been studied for its relationship with oncogenesis. Methylation of DNA, generally at cytosine-phosphate-guanine dinucleotides (CpG) islands, represses transcriptional activity in that region (Figure 2A) [54]. In cancers, CpG site methylation can lead to tumor suppressor genes becoming completely silenced [55]. In breast cancer, abnormal DNA methylation has been observed in a variety of genes, some of which have been identified as diagnostic and prognostic markers [56]. The process and function of gene hypermethylation and hypomethylation in breast cancer metastasis are illustrated in Figure 2B [57]. Agrawal et al. summarized the methylated genes in breast cancers, and de Ruijter et al. reviewed prognostic DNA methylation markers for HR-positive cancer [58,59]. Based on their reports, we summarized the important abnormally methylated (hypo- or hyper-) genes in HR-positive breast cancer and also described their functions in Table 3.

Histones are basic proteins that function to compress DNA within the nucleus to form chromatin, providing a platform for regulating gene transcription. Histone modification (mainly histone acetylation) can occur as a consequence or independent of DNA methylation and provides another mechanism for epigenetic regulation of gene transcription (Figure 2C) [54,60]. The acetylation reaction of histones is controlled by the enzymes histone acetyltransferase (HATs) and histone deacetylases (HDACs). HATs catalyze the transfer of acetyl groups to lysine or arginine residues in histone tails, resulting in gene activation. In contrast, HDACs remove the acetyl groups from histones, resulting in gene repression [61]. The schematic process of histone modification in breast cancer is illustrated in Figure 2D [54]. We will focus on discussing the role of histone modification (HATs and HDACs) in HR-positive breast cancer in this review paper. The studies on HATs and HAT inhibitors in HR-positive breast cancer are not as extensive as those on histone methylation and HDACs and in other subtypes (e.g., basal-like). Peng et al. reported that the widely used Chinese medicine andrographolide inhibited breast cancer viability and proliferation via the inactivation of p300 (a type of HATs) and p-300 mediated acetylation of NF-κB signaling in multiple types of cancer cells, including the HR-positive MCF-7 cells [62]. Another major category related to histone modification is HDACs, whose roles in HR-positive breast cancers have been explored. For example, HDAC1 triggers the proliferation and migration of breast cancer cells (including HR-positive subtypes) via the upregulation of interleukin-8 [63]. HDAC2 was found to be critical in increasing the motility of MCF-7 breast cancer cells via the induction of metastatic markers such as MMP2 and N-cadherin [64]. HDAC3 phosphorylation, mediated by EGFR and c-Src, promoted the invasion of breast cancer cells, including MCF-7 cells [65]. The inhibition of HDAC3 and HDAC6 promoted HDAC inhibitor-induced autophagy and viability reduction in breast cancer cells, including MCF-7 cells [66]. HDACs, such as HDAC2, 4, and 5, have been shown to play a positive role in enhancing tumor progression and drug resistance in HR-positive breast cancer, which is related to their functions in regulating cell proliferation, differentiation, and autophagy [67,68,69,70].

## 3. Gene Expression-Based Assays in Breast Cancer

Currently, a variety of assays and internet tools are available to enhance patient care for breast cancer [71]. Gene expression-based assays are a category of new technologies that identify genes that can be used as a molecular signature in predicting prognosis and guiding therapy. These assays are not only helpful in predicting clinical outcomes but also in making adjuvant chemotherapy decisions. Currently, four main assays, including MammaPrint, Oncotype DX, Breast cancer index, PAM50, and EndoPredict Test, are used to determine recurrence-free survival and/or the use of adjuvant drug therapy in the US. The summary of these assays is listed in Table 4, and each of these assays is described briefly below.

MammaPrint, invented by Agendia, is a genomic test analyzing 70 of the most important genes associated with breast cancer recurrence. It is used to define prognostic outcomes for patients with newly diagnosed invasive breast cancer. The assay can be used for patients of all ages at stage I, II, or operable III, with negative or 1–3 positive lymph nodes and a tumor size up to 5 cm. The test result classifies patients into “low risk” or “high risk” categories in which “low risk” means a low chance of recurrence. “Low-risk” patients are predicted to not benefit from chemotherapy, while “high-risk” patients will have an increased chance of recurrence; thus, adding chemotherapy would result in a better outcome [72].

Oncotype DX is a widely used prognostic and predictive 21-gene qRT-PCR-based assay. It provides breast recurrence scores, which help to predict the chance of metastasis and the likelihood of benefits from chemotherapy for early-stage breast cancer patients using the following criteria: ER-positive (and will be treated with hormone therapy), HER2 negative, negative or 1–3 positive lymph nodes, and a tumor size smaller than 5 cm [73,74,75]. It is also a tumor-profiling test currently used in staging ER-positive, lymph node-negative breast cancer [76].

Prediction Analysis of Microarray 50 (PAM50), invented by Prosigna, tests a group of 50 genes. It helps to predict the chance of metastasis for postmenopausal breast cancer patients with the following criteria: ER-positive (and will be treated with hormone therapy), HER2-negative, negative or 1–3 positive lymph nodes, and a tumor size no larger than 5 cm. A low PAM50 score suggests a low risk of metastasis, and the use of hormone therapy alone may be considered. A high PAM50 score suggests a high risk of metastasis; thus, a more aggressive treatment plan, including both hormone therapy and chemotherapy, will be advised [77,78]. PAM50 is also currently being investigated for its potential use in identifying ER-positive breast cancer patients who may benefit from hormone therapy after 5 years of treatment [79].

Breast Cancer Index (BCI) is the first and only test recognized by the US National Comprehensive Cancer Network (NCCN) for the prediction of extended endocrine therapy benefits in early-stage, HR-positive breast cancer that is either negative or has 1–3 positive lymph nodes [66]. It evaluates the expression of 11 genes and provides two scores. The first score is the BCI prognostic result, which gives the estimate of the likelihood of recurrence within 5–10 years, given as a percentage. The second score is the predictive result, which is given as a “yes” or “no” value and answers whether or not the patient will benefit from an additional 5 years of endocrine therapy [80].

EndoPredict Test, offered by Myriad Genetics, Inc. and available in the US since 2017, is a genomic test for newly diagnosed patients with early-stage, ER-positive, HER2-negative breast cancer (lymph node-negative or node-positive (1–3 nodes), pre- or postmenopausal). The EndoPredict test provides a risk score that is either low risk or high risk of breast cancer recurring as distant metastasis. Knowing if the cancer has a high or low risk of recurrence can help make treatment decisions to reduce risk after surgery. More specially, the test includes proliferation and hormone receptor-related genes that contribute to an accurate assessment of early and late recurrence risk. It was trained and validated on 10-year outcomes data, which offers powerful, 10-year prognostic information for both node-negative and node-positive patients [81,82,83].

## 4. New Drugs and Technology Used in Breast Cancer Diagnosis and Treatment

### 4.1. New Drugs and Drug Targets Used in HR-Positive Breast Cancer

Since most hormone-dependent breast cancers develop resistance to endocrine therapy over time, other pharmaceutical agents with different molecular targets other than estrogen synthesis and receptors have been explored as potential therapy options. Two recent review papers described well the potential new drug targets for the treatment of HR-positive breast cancer, especially to overcome resistance [84,85]. Those drugs include inhibitors of certain signal transduction pathways or molecules (e.g., PI3K/AKT/mTOR, AKT), cell cycle (e.g., CDKs), apoptosis (e.g., Bcl-2), epigenetic changes (e.g., HDAC, NNMT), and immune checkpoints (e.g., PD-1, PD-L1), with some of the drugs having been approved or being investigated in clinical trials. We will briefly discuss CDK inhibitors and epigenetically targeted drugs as examples in this review article.

CDKs are serine/threonine kinases that are the key enzymes involved in regulating cell proliferation through cell-cycle checkpoints via interacting with cyclins. Inactivation of p16 (a tumor suppressor gene and cyclin-dependent kinase inhibitor) and the amplification of the Rb/CDK4/CDK6/cyclin D pathways present in ~50% of primary HR-positive breast cancers, driving cancer cell growth and contributing to endocrine resistance [86]. Adding CDK4/6 inhibitors, such as abemaciclib, palbociclib, or ribociclib, to endocrine therapies showed effectiveness in improving progression-free survival and overall survival in several clinical studies [87,88,89]. In 2021, abemaciclib (Verzenio) was approved by the US FDA as a combinatorial agent used with endocrine therapy, such as tamoxifen or Ais, to treat HR-positive breast cancer. Palbociclib (Ibrance) was approved in combination with fulvestrant and ribociclib (Kisqali) with AIs in 2016 and 2018, respectively. More specially, CDK4/6 inhibitors and the mTOR inhibitor everolimus have been shown to be effective in overcoming the resistance to endocrine therapy caused by *PIK3CA* mutation [90,91]. Although the use of CDK4/6 inhibitors has resulted in progress in treating luminal breast cancer, drug resistance is still an emerging cause of cancer-related death; therefore, overcoming resistance has been an urgent issue to be addressed. The review paper by Li et al. discussed the multiple molecular resistance mechanisms of CDK4/6 inhibitors, such as gene amplification (e.g., ESR1, CDK4, CDK6, p16), pathway activation (e.g., cyclinD1-CDK4/6-Rb, PI3K-AKT-mTOR), and epigenetic alterations [83]. Solutions have been sought to overcome the resistance of CDK4/6 inhibitors. For example, elacestrant, an investigational SERD, has shown growth inhibition in cells resistant to the FDA-approved CDK4/6 inhibitors [92]. Alpelisib, a PI3K inhibitor, was approved by the US FDA in 2019 for use in combination with the fulvestrant to treat postmenopausal women and in men with HR-positive, HER2-negative, PIK3CA-mutated, advanced or metastatic breast cancer following progression on or after an endocrine-based therapy. Alpelisib was able to maintain the sensitivity of CDK4/6 inhibitors in a xenograft mouse model. In the same model, the combination of a PI3K inhibitor, CDK4/6 inhibitor, and endocrine therapy agent further delayed resistance against endocrine therapy and CDK4/6 inhibitors [93]. Similar to PI3K inhibitors, the AKT inhibitor capivasertib, although not officially approved yet, has been investigated in treating HR-positive breast cancer. Combining capivasertib with fulvestrant and palbociclib prevents the progression of breast cancer resistance to CDK4/6 inhibitors and endocrine therapy [94].

As discussed in Section 2.2, epigenetic modifications such as DNA methylation and histone acetylation play a role in tumorigenesis. New agents targeting these processes are being investigated in the prevention and treatment of breast cancer. Hypomethylating agents, such as azacitidine and decitabine, have been approved to treat hematological malignancies, such as myelodysplastic syndromes and acute myeloid leukemia, but not for breast cancer yet. Those drugs have been tested and have been shown to be effective in a variety of breast cancer subtypes, including HR-positive [6,95,96]. Histone deacetylase inhibitors (HDACi) have been approved by the US FDA to treat several cancers, including T-cell lymphoma and multiple myeloma, but not yet for breast cancer. The HDACi abexinostat, which is currently being tested in a US clinical trial to treat follicular lymphoma, was able to restore sensitivity to endocrine therapy via apoptosis induction in tamoxifen-resistant HR-positive MCF7 and T47D cell models [97]. Another clinical trial in the US showed that combining the HDACi entinostat with an AI could restore the sensitivity to endocrine therapy in HR-positive breast cancer [98]. Although some studies of therapy targeting epigenetics in breast cancer showed promising results, disappointing and challenging findings were also reported. For example, a phase II clinical trial of the combination epigenetic therapy of azacitidine and entinostat did not show better clinical outcomes in triple-negative breast cancer patients [99]. The general challenge of established epigenetic drugs is a lack of target specificity. The currently available epigenetic drugs have a global effect on the epigenome rather than on specific targets, causing a global hypomethylation with unwanted effects on oncogenesis and cytotoxicity [100].

### 4.2. New Uses of Technology in the Detection and Treatment of Breast Cancer

In order to improve the early diagnosis and treatment of breast cancer, which eventually benefits the prognosis and survival of patients, novel or new uses of technology have been applied or are being developed. For example, well-known technologies, such as next-generation sequencing, the CRISPR-Cas9 system, artificial intelligence, and CAR T-cell therapy, have been widely used or researched in cancer diagnosis and treatment. Since those technologies have been well and extensively described, in this review paper, we will focus on four relatively new uses of technology in breast cancer: Tomosynthesis Mammographic Imaging Screening Trial (TMIST), mass spectrometry (MS), nanoparticle delivery, and proteolysis targeting chimera (PROTAC). 

Tomosynthesis, also known as 3D mammography, is a new type of digital X-ray mammogram that can create 3D-like images of the breast. The Tomosynthesis Mammographic Imaging Screening Trial (TMIST) is a randomized controlled clinical trial that seeks to prove that the new 3D tomosynthesis mammography is better than the standard 2D digital mammography in reducing advanced breast cancer development in females [101]. Tomosynthesis mammography was proved to be more specific, although not as sensitive, when compared to MRI in breast cancer screening. Goh et al. also suggested that the concurrent use of ultrasound and tomosynthesis mammography provides a more accurate diagnosis [102]. Tomosynthesis mammography allows multiple images to be taken simultaneously and has been shown to overcome some drawbacks caused by traditional 2D mammography, including uncomfortable breast compression and compression-induced missed masses [103].

Mass spectrometry (MS), an analytical technique used to measure the mass-to-charge ratio of ions, remains a reliable approach in the detection of cancer biomarkers and metabolites that are distinguished from normal tissues [104]. For example, MS-based proteomics can be used to identify and measure certain proteins, such as INPP4B, CDK1, and ERBB2, which are associated with breast cancer ER status, tumor grade status, and HER2 status [105]. More and more recent studies have reported the use of MS-related techniques to identify and analyze important proteins or markers in breast cancer. For example, Al-wejeeh et al. performed a comparative proteomic analysis of the different stages of breast cancer tissues using LC-MS-MS [106]. Gawin et al. and Theriault et al., respectively, used MS imaging to reveal intratumor heterogeneity and the metabolomics patterns of breast cancer [107,108]. Single-cell MS imaging was used to identify and visualize cell subtype distribution at the single-cell level in breast cancer, enabling precise and rapid disease diagnosis and prognosis [109]. Thus, the information obtained through MS is not only useful for the identification of tumor subtypes but also helpful for the development of individualized treatment plans [95,110].

Another technology that contributes to breast cancer diagnosis and treatment is nanoparticle delivery. This method ensures drug or contrast media can be precisely delivered to tumor cells, thus limiting the exposure of non-cancer cells, decreasing systemic toxicity, and increasing the half-life of drug and contrast media. Angiogenesis surrounding the tumor usually yields more permeable vasculature, allowing nanoparticles to be delivered to the tumor while sparing the normal tissues [111]. Utilizing nanoparticle delivery has been shown to be useful in treating all subtypes of breast cancer, even in cases with metastasis. For example, Pourtau et al. found that conjugating the HER2-targeted therapy drug trastuzumab (Herceptin) to nanoparticles can specifically target metastasis of the bone [112]. It has also been proven that the nanoparticle delivery system can overcome drug resistance in certain patients by evading drug efflux pumps [113]. Additionally, hematopoietic stem cells that are nonimmunogenic and that are able to bypass immune surveillance can be used as drug carriers to ensure the delivery to target sites in tumors [114]. Besides delivering drugs, using nanoparticles can also be used to deliver intravenous contrast in MRI, which has been proven to increase imaging sensitivity and the blood circulation time of the contrast agent [111]. More recently, Guo et al. developed an antibody-conjugated tumor-targeted nanolipogel that can precisely deliver CRISPR/Cas9 plasmids into triple-negative breast cancer for genomic editing [115]. Liu et al. used the lipid nanoparticles to deliver protein kinase N3 (PKN3, overexpressed in breast cancer cells) shRNA in a mouse model and found that the system had a PKN3 inhibition rate of 60.8% and tumor inhibition rate of 62.3%, indicating a potential therapeutic strategy for breast cancer [116]. In addition, nanomedicine-based immunotherapy may be a viable option for treating breast cancer as well [117].

The proteolysis targeting chimera (PROTAC) technology has been developed for targeted protein degradation through the ubiquitin-proteasome system, with potential use for cancer therapy. The PROTAC molecule generally consists of a ligand (normally small-molecule inhibitor) from the protein of interest and a covalently linked ligand of an E3 ubiquitin ligase (E3). Upon binding to the targeted protein, the PROTAC can recruit E3 for ubiquitination, which is subjected to proteasome-mediated degradation [118,119]. PROTAC is being developed to target clinically validated proteins in cancer therapy, such as Von Hippel-Lindau, methionyl aminopeptidase 2, and BCL6. Specifically, the oral PROTAC ARV-471 that targets ER is currently in a phase 1/2 trial for the treatment of ER-positive advanced or metastatic breast cancer with or without combination therapy with palbociclib [120].

In summary, in this review article, we focused on the important features of HR-positive breast cancer, including HR structure and signaling, genetics, including epigenetics and gene mutations, gene expression-based assays, drug treatments, and new uses of technology. This hopefully highlighted the importance of individualized therapies to maximize patient care.

## Figures and Tables

**Figure 2 cancers-15-01303-f002:**
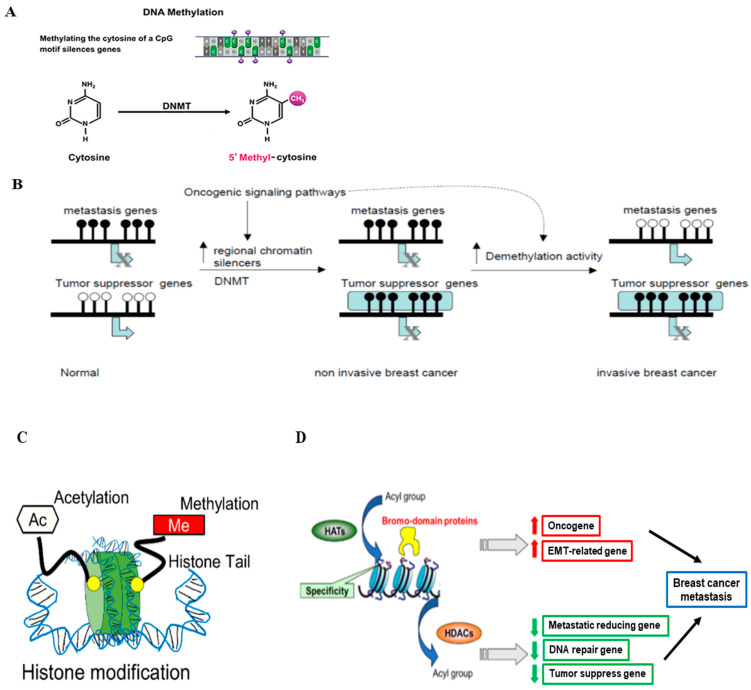
Epigenetic regulation in breast cancer. (**A**) The schematic process of DNA methylation. DNA methylation is mediated by DNA methyltransferases (DNMTs) (Figure modified from [54]). (**B**) Hypermethylation and hypomethylation in breast cancer. In normal epithelial breast cells, tumor suppressor genes are active and unmethylated, while genes required for metastasis are methylated and inactive. Activation of oncogenic pathways leads to the interaction of specific repressors with certain tumor suppressors leading to chromatin inactivation and DNA methylation (Figure reprinted/adapted with permission from Ref. [57]. Copyright 2004 Elsevier Inc.). (**C**) The schematic representation of histone modifications that occur in breast cancer progression, such as histone methylation and histone acetylation (Figure reproduced from [54]). (**D**) The proposed role of histone modification in the transcriptional regulation of genes involved in breast cancer metastasis and related functions (Figure reproduced from [54]).

**Table 1 cancers-15-01303-t001:** Breast cancer subtypes based on gene expression profiling.

Subtype	Gene Profiling	Characters
Luminal A	ER+ and/or PR+, HER2−, low Ki67 index	Most common, ~50–60%, low grade and proliferation, good prognosis, low relapse rate
Luminal B	ER+ and/or PR+, HER2+ or HER2-, high Ki67 index	~15–20%, higher proliferation and worse prognosis compared to luminal A
HER2 positive (HER2 enriched)	HER2+, ER-, PR-	~15–20%, aggressive, poor prognosis
Basal-like (triple-negative)	ER-, PR-, HER2-	Aggressive, high metastasis, poor prognosis

**Table 2 cancers-15-01303-t002:** Commonly mutated genes in HR-positive breast cancer (alphabetical order).

Gene	Full Name	Function
*AAMDC*	adipogenesis associated Mth938 domain containing	Regulates fat cell differentiation
*AGTR2*	angiotensin II receptor type 2	Receptor for angiotensin II
*AKT1*	AKT serine/threonine kinase 1	Cell growth and division
*ATXN1*	ataxin 1	DNA-binding protein
*ARMCX1*	armadillo repeat-containing X-linked protein 1	Regulates mitochondrial transport during axon regeneration
*BRCA2*	BReast CAncer gene 2	Tumor suppressor gene
*C10orf90*	chromosome 10 open reading frame 90	Enables the activities of histone deacetylase binding, microtubule binding, and ubiquitin ligase
*CA4*	carbonic anhydrase 4	Zinc metalloenzyme that catalyzes the reversible hydration of carbon dioxide
*CBFB*	core-binding factor, beta subunit	Regulates hematopoiesis and osteogenesis
*CDKN1B*	cyclin-dependent kinase inhibitor 1B	Encodes a cyclin-dependent kinase inhibitor
*CHD1*	cadherin-1	Tumor suppressor gene, makes epithelial cadherin (E-caderin)
*CHST6*	carbohydrate sulfotransferase 6	Produces keratan sulfate
*DDB1*	damage-specific DNA binding protein 1	Regulates nucleotide excision repair, a core component of the CUL4A- and CUL4B-based E3 ubiquitin ligase complexes
*DPH2*	diphthamide biosynthesis 2	Regulates diphthamide synthesis
*ESR1*	Estrogen Receptor 1	Encodes ERα, associated diseases include estrogen resistance and breast cancer
*FAM48A*	family with sequence similarity 48, member A	Transcription coregulator activity
*FAM91A1*	family with sequence similarity 91, member A1	Facilitates the golgin-mediated capture of vesicles
*FOXF1*	forkhead box F1	Transcription factor, important in the development of pulmonary mesenchyme and gastrointestinal tract
*FOXP1*	forkhead box P1	Transcription factor, important in the development of brain, heart, and lung
*GATA3*	GATA binding protein 3	Transcription factor, important in tissue development and immune responses
*GSS*	glutathione synthetase	Participates in gamma-glutamyl cycle
*HDLBP*	high-density lipoprotein binding protein (vigilin)	Binds high-density lipoprotein and regulates cholesterol levels in cells
*HEG1*	heart development protein with EGF-like domains 1	Calcium ion binding activity, involved in cell-cell junction assembly and heart development
*KCNH4*	potassium voltage-gated channel, subfamily H member 4	Voltage-gated potassium channel
*KIAA1522*	KIAA1522	Cell differentiation
*KIF26B*	kinesin family member 26B	Intracellular motor protein that transports organelles along microtubules
*KLK10*	kallikrein related peptidase 10	Serine protease
*KNDC1*	kinase non-catalytic C-lobe domain containing 1	Ras guanine nucleotide exchange factor
*LCE3E*	late cornified envelope 3E	Keratinization
*LDLRAP1*	low-density lipoprotein receptor adaptor protein 1	Interacts with the low-density lipoprotein receptor
*LPAR1*	lysophosphatidic acid receptor 1	Reorganization of the actin cytoskeleton, cell migration, differentiation, and proliferation
*LRRTM4*	leucine-rich repeat transmembrane neuronal 4	Enables heparan sulfate proteoglycan binding activity, regulates synapse assembly
*MAP3K1*	mitogen-activated protein kinase kinase kinase 1	Serine/threonine kinase involved in signal transduction pathways
*MLL3* (also known as *KMT2C*)	myeloid/lymphoid or mixed-lineage leukemia 3 (lysine N-methyltransferase 2C)	Encodes a nuclear protein, possesses histone methylation activity, involved in transcriptional coactivation
*MRPS34*	mitochondrial ribosomal protein S34	Regulates protein synthesis within mitochondria
*MUC2*	mucin 2, oligomeric mucus/gel-forming	Codes mucin protein which secrets and forms an insoluble mucous barrier that protects the gut lumen
*MUC4*	mucin 4, cell surface associated	A major constituent of mucus
*NIN*	ninein	Regulates centrosomal function, positions and anchors microtubules minus-ends in epithelial cells
*MYH9*	myosin heavy chain 9	Encodes a conventional non-muscle myosin
*MYO1A*	myosin IA	Encodes a member of the myosin superfamily, which functions as actin-based molecular motors
*NPHP3*	nephronophthisis 3	Renal tubular development and function
*NRXN3*	neurexin 3	Function in the nervous system as receptors and cell adhesion
*OR5AK2*	olfactory receptor, family 5, subfamily AK member 2	Encodes for olfactory receptor proteins
*PCDH11X*	protocadherin 11 X-linked	Cell–cell recognition essential for segmental development and function of central nervous system
*PCGF2*	polycomb group ring finger 2	Cell proliferation, neural cell development
*PIGW*	phosphatidylinositol glycan anchor biosynthesis class W	Encodes inositol acyltransferase that acylates the inositol ring of phosphatidylinositol
*PIK3CA*	phosphoinositide-3-kinase, catalytic, alpha polypeptide	Encodes the catalytic subunit of phosphatidylinositol 3-kinase
*POMK* (also known as *SGK196*)	protein O-mannose kinase	Regulates the formation of transmembrane linkages between the extracellular matrix and exoskeleton
*PPFIBP2*	PPFIA binding protein 2	Axon guidance and neuronal synapse development
*PRDM14*	PR domain containing 14	Encoded protein has histone methyltransferase activity and regulates cell pluripotency
*PRG3*	proteoglycan 3	Extracellular matrix structural constituent
*PRSS36*	protease, serine, 36	Enables serine-type endopeptidase activity
*PTEN*	phosphatase and tensin homolog	Tumor suppressor gene
*RB1*	RB transcriptional corepressor 1	Tumor suppressor gene
*RESF1*	Retroelement silencing factor 1	Enable activities of histone binding and histone methyltransferase binding
*RUNX1*	RUNX family transcription factor 1	Encodes core binding factor which is a transcription factor
*SF3B1*	splicing factor 3b subunit 1	Encodes subunit 1 of the splicing factor 3b protein complex
*SLC13A3*	solute carrier family 13, member 3	Induces sodium-dependent inward currents in the presence of succinate and dimethylsuccinate
*SLC38A8*	solute carrier family 38, member 8	Encodes a putative sodium-dependent amino-acid/proton antiporter
*STMN2*	stathmin 2	Encodes a member of the stathmin family of phosphoproteins which function in microtubule dynamics and signal transduction.
*TAF15*	TATA-box binding protein associated factor 15	Regulates RNA polymerase II gene transcription
*TBX3*	T-Box transcription factor 3	Encodes transcription factor
*TET3*	Tet methylcytosine dioxygenase 3	DNA demethylation
*TOB1*	transducer of ERBB2, 1	Encodes anti-proliferative factors which regulate cell growth
*TP53*	tumor protein 53	Tumor suppressor gene
*TRIM61*	tripartite motif containing 61	Enables ubiquitin ligase activity
*ZBED4*	zinc finger, BED-type containing 4	Enables identical protein binding activity, regulates transcription by RNA polymerase II
*ZC3HC1*	zinc finger, C3HC-type containing 1	Encodes an F-box-containing protein that is a component of an SCF-type E3 ubiquitin ligase complex that regulates the onset of cell division
*ZNF345*	zinc finger protein 345	Enables sequence-specific double-stranded DNA binding activity, regulates transcription by RNA polymerase II

**Table 3 cancers-15-01303-t003:** Common abnormally methylated genes in HR-positive breast cancer (alphabetical order).

Hypermethylated Gene	Full Name	Function
*14-3-3* σ	*14-3-3* σ	Cell cycle regulator
*AK5*	adenylate kinase 5	Adenylate kinase
*AMN*	amnion-associated transmembrane protein	Regulates bone morphogenetic protein receptor function
*APC*	adenomatous polyposis coli	Tumor suppressor gene
*BRCA1*	breast cancer 1	Tumor suppressor gene
*CCND2*	cyclin D2	Cell cycle regulator
*CDH1*	cadherin 1	Encodes E-cadherin (epithelial marker)
*CDH13*	cadherin 13	Encodes a cadherin superfamily
*CDKN1C* (also known as *p57KIP2*)	cyclin-dependent kinase inhibitor 1C	Cell cycle regulator
*CDKN2A*	cyclin-dependent kinase inhibitor 2A	Cell cycle regulator
*ESR1*	estrogen receptor 1	Encodes estrogen receptor
*GSTP1*	glutathione S-transferase P	Detoxification
*FOXA2*	forkhead box A2	DNA-binding protein and transcriptional activator
*GJB2*	gap junction protein beta 2 (also known as connexin 26)	Encodes gap junction protein
*GSTP1*	glutathione S-transferase pi	Detoxification
*HOXD11*	Home box D11	Transcription factor
*ID4*	inhibitor of DNA binding 4	Regulates prenatal development and tumorigenesis
*LINE-1*	long interspersed nuclear element-1	Transposable element
*p14ARF*	p14 alternate reading frame	Tumor suppressor gene
*p16INK4a*	p16 INK 4a	Tumor suppressor gene
*PCDH10*	protocadherin 10	Encodes a cadherin superfamily
*PGR*	progesterone receptor	Encodes progesterone receptor
*PITX2*	paired like homeodomain 2	Transcription factor
*PTPRO*	protein tyrosine phosphatase receptor type O	Encodes receptor-type protein tyrosine phosphatase
*Rad9*	Rad9 homolog	Cell cycle checkpoint
*RARB*	retinoic acid receptor beta2	Nuclear transcriptional regulators
*RASSF1*	Ras-association domain family protein 1A	Tumor suppressor function
*RASSF5 * (also known as *NORE1*)	Ras association domain family member 5	Tumor suppressor function
*RUNX3*	RUNX family transcription factor 3	Transcription factor
SCGB3A1 (also known as HIN1)	secretoglobin family 3A member 1	Regulates myoblast fusion
*SCL6A20 * (also known as *XT3*)	solute carrier family 6, member 20	Transports small hydrophilic substances across cell membranes
*SFN*	stratifin	Cell cycle checkpoint
*SFRP2*	secreted frizzled-related protein 2	Regulates Wnt signaling
*STK11* (also known as *LKB1*)	serine/threonine kinase 11	Encodes serine/threonine kinase family, regulates cell polarity, and functions as a tumor suppressor
*SIM1*	SIM BHLH transcription factor 1	Transcription factor
synuclein γ	synuclein gamma	Involved in pathogenesis of neurodegenerative diseases
*TPM1*	tropomyosin 1	Involved in the contractile system of striated and smooth muscles and the cytoskeleton of non-muscle cells
*TSPAN-2*	tetraspan 2	Regulates signal transduction
*Twist1*	Twist family BHLH transcription factor 1	Transcription factor
*uPA*	urokinase plasminogen activator	Serine protease involved in degradation of extracellular matrix and tumor cell migration and proliferation
*WT-1*	WT1 transcription factor	Transcription factor
**Hypomethylated Gene**	**Full Name**	**Function**
*ARHI/NOEY2*	aplasia Ras homolog member I	Tumor suppressor gene
*c-MYC*	c-MYC proto-oncogene	Proto-oncogene
*MAGE*	melanoma-associated antigen	Tumor-specific antigen

**Table 4 cancers-15-01303-t004:** Common gene expression-based assays in breast cancer.

Assay Name	Clinical Use	Factors Being Evaluated	Patient Demographics	Scoring
MammaPrint	Prognostic, determines chance of recurrence	70 genes associated with breast cancer recurrence	Newly diagnosed invasive breast cancer patients of all ages, stage I, II, or operable III, negative or 1–3 positive lymph nodes, and tumor size up to 5 cm	“low risk” or “high risk”
Oncotype DX	Predictive and prognostic, predicts recurrence and benefit from chemotherapy, staging	21 genes: 16 cancer related and 5 reference genes	Early-stage, ER-positive, HER2-negative, negative or 1–3 positive lymph nodes, and tumor size smaller than 5 cm	Risk score (≤15, low-risk; 16–25, intermediate-risk; ≥26, high-risk).
PAM50	Predicts chance of metastasis and determines if chemotherapy is needed	50 genes	Postmenopausal women, ER-positive, HER2-negative, negative or 1–3 positive lymph nodes, and tumor size no larger than 5 cm	PAM50-based Prosigna risk of recurrence score: low or high
Breast Cancer Index	Predicts risk of recurrence and mortality in 5 years and whether the benefit of extended endocrine therapy for 5 years will reduce risk of recurrence	11 genes	Early-stage HR-positive, lymph node-negative or positive (1–3 nodes)	Prognostic result (percentage) and predictive result (“yes” or “no”)
EndoPredict Test	Predicts the likelihood of distant recurrence with 10 years after diagnosis	12 genes	Early-stage ER-positive, HER2-negative, lymph node negative or positive (1–3 nodes)	EPclin Risk Score (a number between 1.1 and 6.2) maps to a percentage risk of recurrence. The scores higher than 3.3287 are interpreted as high-risk, and lower than 3.3287 are interpreted as low-risk of recurrence.

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
