# Peer review of "Genetics, Treatment, and New Technologies of Hormone Receptor-Positive Breast Cancer"

_cancers, 2023, doi:10.3390/cancers15041303_

Round 1

Reviewer 1 Report

This is yet another review on hormone receptor - positive breast cancer. 

Simple Summary: They included in this review 'novel and new uses of technology used in diagnosis and treatment'.  I am not so sure that this part of the manuscript is so useful for the reader. Also, to what extent should we use or being updated on 'epigeetics' and 'abnormally methylated genes' in treating HR-positive breast cancer in our current daily practice?

Line 53: How should we interprete 'smoking' as a breast cancer risk factor which in the literature is controversial. Smoking induces an earlier menopause which might lead to less breast cancer. There is some evidence that those who started to smoke at adolescent or peri-menarcheal ages have more premenopausal breast cancer. Obesitas protects against premenopausal breast cancer. The authors don't even mention 'alcohol' which is a well known breast cancer risk factor!

Line 55: Classification of breast cancer can be based on anatomical localisation... I am not sure what they mean by this but that is an incorrect statement

Line 57-62: Important to separate pathological classification of breast cancer based on immunohistochemistry and molecular classification using PAM50 (or Blue Print). If they suggest to use Ki-67 for BC classification they should be critical as there is no consensus how to differentiate luminal A from luminal B. Also, HER2- positive HR-pos breast cancers are different from those not over-expressing HER2.

Note: Hormone receptor should be replaced by estrogen as 'other' hormone receptors are also expressed (androgen, glucocorticoid, ...). 

Line 62: Please indicate the 'exact' target in HR-pos breast cancer?

Line 75: Twice they state 'estradiol'

Page 3: Please clearly indicate text to go with Fig 1. Please avoid repetition between text manuscript and text to go with Fig. How important is the storu of ER-beta which we don't use in the clinic for BC treatment? Also, the exact role of ER-beta remains obscure and I am not convinced after reading the manuscript it plays an important role!

Line 111: there is a word missing in this sentence

Line 127 and others: @ avoid this sign appearing in the text I did receive (to differentiate between alfa and beta. 

Line 144: How essential/important is the progesterone receptor for the regulation of ovulation?

Line 162: Progesterone is important for the PR to be expressed. I don't follow this. Male breast cancer is mostly PR-pos and there is no circulating progesterone. 

Line 164:The authors state there are over time less PR-neg HR-pos breast cancers as immunohistochemical techniques are being optimized.  I am not convinced we have less PR-pos cases.

Note: I miss the importance of quantitative ER-expression in the pathological work up. qER is predictive and therefore prognostic. Also qPR is important to mention!

Line 193: There is evidence that premenopausal women benefit from AI if their ovarian function is suppressed. So the word 'might' is strange!

Line 200: This reference concerns a retrospective subanalysis of 2 randomized clincal trials and hypotheses generating. Please also mention how tamoxifen is effective in case of ER-mutations!

Line 213: Endocrinetherapy resistant mechanisms to breast cancers--> do they mean early or metastatic HR-pos breast cancers? Some have observed PIK3CAm not to be predictive for endocrine therapy (adjuvant/metastatic)! There is inconsistency in the recent literature and the authors shoud discuss this more deeper.

Line 247 + 284: The information in the table should be resitricted to what is important for our daily clinical practice! This now is too extensive!

Line 270: Please clarify the meaning of 'metastatic' genes

Line 337: The authors are convinced that ODx is the only tumor profiling test currently used in staging ER-pos LN-neg breast cancer. The word staging is strange. I am not convinced that sentence is correct. Many of clinicians use MammaPrint. There is enough evidence to use other tests than Oncotype DX to add adjuvant chemotherapy.

Table 4. What is a breast cancer has a RS of '15'?

Line 388. If PIK3CA is mutated, CDK 4/6 inhibors can be active. They are not predictive. Please correct.

Note: Also discuss the role of mTOR inhibition with everolimus (if PIK3CA is mutated).

Line 404: The authors discuss agents targeting epigenetic modifications. Please update with more recent (disappointing) findings.  

Line 422: Robotic surgery: Please clarify what they mean?

Part 2. New uses of technology... Why do they select these 4 specific topics and not for example anti-body drug conjugates or artificial intelligance?? Why to discuss mass spectrometry while there is no clinical application

Line 436: This paragraph is not well written and particular that sentence confusing. Do they mean screening?

Line 446 and rest of paragraph: Not of any use for a clinician treating breast cancer. Also, this information is not up-to-date?

Author Response

Please also see the attachment (word document).

Authors’ Response to Reviewers’ Comments 

We appreciate the Editor and reviewers spending great time to read our manuscript carefully with excellent suggestions. Please see our responses to each comment as follow.

Respond to reviewer #1:

This is yet another review on hormone receptor - positive breast cancer.

Simple Summary: They included in this review 'novel and new uses of technology used in diagnosis and treatment'.  I am not so sure that this part of the manuscript is so useful for the reader. Also, to what extent should we use or being updated on 'epigenetics' and 'abnormally methylated genes' in treating HR-positive breast cancer in our current daily practice?

Response: We appreciate the excellent comment from the reviewer. Our manuscript was invited by the Special Issue “Personalized Therapy for Hormone-Responsive Breast Cancer”, thus it is not surprising that there are multiple manuscripts on this topic.

Regarding the new uses of technology section, as we have mentioned in the manuscript, well-known technologies used in breast cancer such as next-generation sequencing (discussed in the “gene mutation in HR-positive breast cancer” section, and PMID: 23989017, PMID: 23879964, PMID: 30689231), the CRISPR-Cas9 system (PMID: 35879772, PMID: 33031826, PMID: 33926574), artificial intelligence (PMID: 35884503, PMID: 35771379, PMID: 35623118), CAR T-cell therapy (PMID: 35414783, PMID: 32795486, PMID: 33442419), and robotic surgery (PMID: 35815331, PMID: 32095394) have been well and extensively described in many other review papers. Thus, we focused on other four relatively new uses of technology (TIMIST, mass spectrometry, nanoparticle delivery, and PROTAC) in breast cancer in our manuscript.

Regarding the comment on “epigenetics” and “abnormally methylated genes”, we think it is important to discuss in the manuscript. Epigenetic regulation of gene expression plays an important role in breast cancer development and progression. Thus, understanding epigenetics in breast cancer is crucial to reveal the molecular and genetic mechanisms of cancer development. Many new drugs or therapies are or being developed based on these epigenetic targets. The readers of the “Cancers” journal not only include clinical physicians, but also include basic scientists and researchers. With the discussion of cancer genetics related topics, our review paper will be beneficial for health care professionals and researchers to have a better understanding of the basic science of breast cancer. In addition, to our knowledge, there is no such a review or research paper specifically listed and summarized the commonly mutated and abnormally methylated genes in HR-positive breast cancer, thus, as highlighted in the abstract, we think this is a strength of our review paper.  

Line 53: How should we interpret 'smoking' as a breast cancer risk factor which in the literature is controversial. Smoking induces an earlier menopause which might lead to less breast cancer. There is some evidence that those who started to smoke at adolescent or peri-menarcheal ages have more premenopausal breast cancer. Obesitas protects against premenopausal breast cancer. The authors don't even mention 'alcohol' which is a well known breast cancer risk factor!

Response: Thanks for the excellent comments. We have added a reference and revised our manuscript accordingly based on the comments.

We agree with the editor that the precise role of smoking in breast cancer is not fully known. However, smoking as a risk of breast cancer and some other cancer types has been described in multiple publications and studies (e.g., https://www.breastcancer.org/risk/risk-factors/smoking, Jones et al., Breast Cancer Research, 2017, 19:118, PMID: 29162146). The paper concluded that “smoking was associated with a modest but significantly increased risk of breast cancer, particularly among women who started smoking at adolescent or peri-menarcheal ages. The relative risk of breast cancer associated with smoking was greater for women with a family history of the disease”.

Regarding obesity, the U.S. Centers of Disease Control and Prevention (CDC) listed “being overweight or having obesity after menopause” as a risk factor for breast cancer (https://www.cdc.gov/cancer/breast/basic_info/risk_factors.htm). We have changed from “obesity” to “obesity after menopause” in our revised manuscript.

          We have added “drinking alcohol” as a risk in our revised manuscript.

Line 55: Classification of breast cancer can be based on anatomical localization. I am not sure what they mean by this but that is an incorrect statement.

Response: Sorry about the confusion and we have revised it in our manuscript. It should be clear now.  

There are different ways to describe the types or classification of breast cancer. Our manuscript mainly discussed the molecular receptor classification (Table 1). Other classification can be based on anatomical localization, for example, from nipple to stroma, there are different types of breast cancer, such as Paget’s disease, ductal cancer, tubular cancer, and phyllodes tumor. Furthermore, malignant breast tumor can also be pathologically classified such as noninvasive (in situ) or invasive cancer. For the reviewer’s reference, I, the corresponding author, attached two slides from my breast cancer lecture taught to medical students (I am a tenured faculty member in a U.S. medical school). The related information can also be found at the American Cancer Society website: https://www.cancer.org/cancer/breast-cancer/about/types-of-breast-cancer.html. We think the additional information we provided can better explain what “anatomical localization” means. 

Line 57-62: Important to separate pathological classification of breast cancer based on immunohistochemistry and molecular classification using PAM50 (or Blue Print). If they suggest to use Ki-67 for BC classification they should be critical as there is no consensus how to differentiate luminal A from luminal B. Also, HER2- positive HR-pos breast cancers are different from those not over-expressing HER2.

Response: Thanks for the excellent comment. We described the classification based on the references we cited. Please also see Table 3 in the paper (Eliyatkın et al., J Breast Health, 2015, 11(2):59-66, PMID: 28331693).

To be more precise, we have added the reference and revised our manuscript based on the comment. We added “However, the appropriateness of using Ki-67 as a proliferation marker has been questioned. The application of a Ki67 scoring as positive/negative or high/low in patient follow-up and treatment is controversial and there is no consensus on this issue today” in our revised manuscript.

Note: Hormone receptor should be replaced by estrogen as 'other' hormone receptors are also expressed (androgen, glucocorticoid, ...).

Response: We agree with the reviewer that as a general concept, hormone is a broad term which includes estrogen, androgen, cortisol, insulin, thyroid hormone, etc. However, specifically to breast cancer, hormone receptor commonly refers to estrogen and progesterone receptors, which is well accepted. Thus, we did not make changes on this.   

Line 62: Please indicate the 'exact' target in HR-pos breast cancer?

Response: Thanks for the excellent comment. “Exact” means that different drugs/therapies can be used based on the subtypes. Based on the comment, we have revised the manuscript to be clearer.

Line 75: Twice they state 'estradiol'.

Response: We have corrected the error in the revised manuscript.

Page 3: Please clearly indicate text to go with Fig 1. Please avoid repetition between text manuscript and text to go with Fig. How important is the storu of ER-beta which we don't use in the clinic for BC treatment? Also, the exact role of ER-beta remains obscure and I am not convinced after reading the manuscript it plays an important role!

Response: Thanks for the excellent comment. We have revised the manuscript based on the comment. We have clearly indicated the text corresponding to Figure 1 and reduced repetition content between text and figure legends in the revised manuscript.

We think the “storu of ER-beta” should be “story of ER-beta”. The description of estrogen and ER subtypes are well described in references #6-12 in the original manuscript, so we did not put very detailed information on them. ER-alpha and ER-beta are both present in breast tissues. Depletion of ER-alpha leads to failure to initiate mammary gland growth and maturation, however, ER-beta knockout has little effects on mammary gland development. Based on the comment, we have added more references and information on ER-alpha and ER-beta in the revised manuscript.

Line 111: there is a word missing in this sentence.

Response: We have revised the sentence based on the comment.

Line 127 and others: @ avoid this sign appearing in the text I did receive (to differentiate between alfa and beta.

Response: We have corrected these errors in the revised manuscript.

Line 144: How essential/important is the progesterone receptor for the regulation of ovulation?

Response: We have added two references (PMID: 10781075, PMID: 32294429) and revised our manuscript based on the comment. We have revised the sentences to “Progesterone, similar to estrogen, is essential for the regulation of normal reproductive functions. In female, progesterone plays a vital role especially in mammary development/maturation and ovulation. In male, progesterone influences spermiogenesis and testosterone production from Leydig cells. The PR has also been shown to play an essential role in triggering ovulation and attenuating ovulatory inflammation.” in the revised manuscript.   

Line 162: Progesterone is important for the PR to be expressed. I don't follow this. Male breast cancer is mostly PR-pos and there is no circulating progesterone.

Response:  Thanks for the comment. Based on the comment, we have deleted the sentence “Estrogen is generally required to induce the expression of PR, thus both hormones are necessary for PR to be expressed.” in the revised manuscript.  

Line 164: The authors state there are over time less PR-neg HR-pos breast cancers as immunohistochemical techniques are being optimized.  I am not convinced we have less PR-pos cases.

Response:  Thanks for the comment. Does the reviewer mean the PR-negative and ER-positive cases, instead of PR-neg and HR-pos?

Our manuscript writes “…only ~1% of breast cancer cases appear to be ER-negative and PR-positive, causing the study of PR to be overlooked. In addition, the number of those cases continually decreases…”.  Our information is based on the references we cited. For example, in the reference #20 [Giannakeas, JAMA Netw Open, 2020, 3(1): p. e1918176] cited in our original manuscript, it says “In the study by Li and colleagues, a finding against the case that the ER-negative/PR-positive subtype is a real category is the decline in proportions from 4.5% to 1.0% from 1990 to 2015. Nowadays, only 1% of breast cancer cases are categorized as ER-negative/PR-positive, and the problem is not as large as it once was. It is unlikely that cancer types are evolving in response to a changing environment”. Thus, we think the information in our manuscript is correct and reflects the current status.

Note: I miss the importance of quantitative ER-expression in the pathological work up. qER is predictive and therefore prognostic. Also qPR is important to mention!

Response:  Thanks for the comment. We have revised our manuscript based on the comment.  We have changed the sentence from “…decreases as immunohistochemical techniques are being optimized.” to “…decreases as molecular biology techniques such as immunohistochemistry and quantitative PCR (qPCR) are being optimized” in our revised manuscript.

Line 193: There is evidence that premenopausal women benefit from AI if their ovarian function is suppressed. So the word 'might' is strange!

Response:  Thanks for the comment. We have changed the word from “might” to “can” in our revised manuscript.

Line 200: This reference concerns a retrospective subanalysis of 2 randomized clinical trials and hypotheses generating. Please also mention how tamoxifen is effective in case of ER-mutations!

Response:  Thanks for the comment. The information is based on the references we cited.

Some reports said that ER mutations are resistant to tamoxifen. For example, In Fuqua et al.’s paper (PMID: 24487689), it says “The Y537N ER mutation exhibited highly elevated ligand-independent, constitutive transcriptional activity; thus cells expressing the mutant were resistant to tamoxifen treatment…”. In Tonetti et al.’s paper (PMID: 9393947), it says “A plethora of studies have reported the detection of estrogen receptor mRNA splice variants, and it has been suggested that the accumulation of these variant mRNAs are responsible for the development of tamoxifen-resistant breast cancer”. Alluri et al.’s paper (PMID: 25928204) reported that tamoxifen may cause ER mutations which led to resistance.

In this paragraph, we mainly focused on discussing SERDs. Thus, we kept the content the same in our manuscript.

Line 213: Endocrine therapy resistant mechanisms to breast cancers--> do they mean early or metastatic HR-pos breast cancers? Some have observed PIK3CA not to be predictive for endocrine therapy (adjuvant/metastatic)! There is inconsistency in the recent literature and the authors should discuss this more deeper.

Response:  Thanks for the comment. Regarding the timeline of resistance occurs, based on Lei et al.’s paper (PMID: 31839155), it says “While most ER+ breast cancer may initially respond to endocrine treatment, 15–20% of tumors are intrinsically resistant to treatment, and another 30–40% acquire resistance to treatment over a period of many years. Resistance to treatment inevitably results in relapse and metastasis, leading to death”. Based on the comment, we have added this reference in our revised manuscript.

In our manuscript, we discussed common genetic mechanisms or gene mutations that cause endocrine therapy resistance, and PI3CA gene mutation is one of them. We agree with the reviewer that PI3KCA may not be a predictive maker for the effectiveness of therapy at this point of time. We do not plan to discuss this in detail in the manuscript.  

Line 247 + 284: The information in the table should be restricted to what is important for our daily clinical practice! This now is too extensive!

Response:  Thanks for the comment. As we have mentioned in “response to simple summary”, regarding the contents of “epigenetics” and “abnormally methylated genes”, we think it is important to keep and discuss. Understanding epigenetics in breast cancer is crucial to reveal the molecular and genetic mechanisms of cancer development. Many new drugs or therapies are or being developed based on these epigenetic targets. As one of the top oncology journals, “Cancers” has broad readers including not only clinical physicians, but also basic scientists and researchers. With the discussion of these topics, our review paper will be beneficial for health care professionals as well as researchers to have a better understanding of the basic science of breast cancer. Based on our knowledge, our paper will be the first one which comprehensively summarized the commonly mutated genes and abnormally methylated genes in HR-positive breast cancer. We think this is a strength of our review paper.  

Line 270: Please clarify the meaning of 'metastatic' genes

Response: Thanks for the comment. “Metastatic genes” means “genes required for metastasis”. It was described like this in the reference #43 cited in our original manuscript. We have revised it based on this comment.

Line 337: The authors are convinced that ODx is the only tumor profiling test currently used in staging ER-pos LN-neg breast cancer. The word staging is strange. I am not convinced that sentence is correct. Many of clinicians use MammaPrint. There is enough evidence to use other tests than Oncotype DX to add adjuvant chemotherapy.

Response: Thanks for the comment. We should add a geographical scope of the approved use of gene expression-based assays. The information described in the manuscript is in the United States (U.S.), the situation may be different in Europe or other countries in the world. We have revised it based on this comment. 

Please see the most recent version of U.S. National Comprehensive Cancer Network (NCCN) guideline (Breast Cancer, Version 4. 2022, https://www.nccn.org/guidelines/guidelines-detail?category=1&id=1419), on page 73 of the 250-page PDF, it clearly says that Oncotype  DX is the only assay for both predictive and prognostic as far. Please see the information in the table. Predictive means clinical outcomes and prognostic means benefit from chemotherapy. Some other assays also have good prognostic estimates in clinical trials, but they are not officially approved yet, based on the latest NCCN guideline information. Thus, the information in our manuscript is correct.    

Regarding the term staging, we think it is a common term used in oncology. Please see more information in the reference #62 in the original manuscript. The paper also says that Oncotype DX recurrent score is helpful in defining the stage. In addition, based on the information from the Komen.org (https://www.komen.org/breast-cancer/diagnosis/factors-that-affect-prognosis/oncotype-dx/#:~:text=Oncotype%20DX%20can%20be%20included,other%20factors%20to%20determine%20stage), it says “Oncotype DX can be included as part of breast cancer staging for some estrogen receptor-positive, lymph node-negative tumors. It’s the only tumor profiling test used in breast cancer staging today. If Oncotype DX testing is done, the results are used in combination with other factors to determine stage”. Thus, we kept the word “stage” the same in the manuscript.

Based on the comment, to be more precise, we have changed from “the only” to “a” in the revised manuscript.

Table 4. What is a breast cancer has a RS of '15'?

Response: Thanks for the comment. Oncotype DX risk score 15 or lower is considered as low risk. We have changed from “Risk score (<15, low-risk)” to “Risk score (≤15, low-risk)” in the revised manuscript. 

Line 388. If PIK3CA is mutated, CDK 4/6 inhibits can be active. They are not predictive. Please correct.

Response: Thanks for the comment. We have revised our manuscript based on the comment.  We have deleted “…mutation (e.g., PIK3CA) and …” in the revised manuscript.

Note: Also discuss the role of mTOR inhibition with everolimus (if PIK3CA is mutated).

Response: Thanks for the comment. Based on the comment, we have added two references and the sentence “More specially, CDK4/6 inhibitors and the mTOR inhibitor everolimus had been shown to be effective to overcome the resistance to endocrine therapy caused by PIK3CA mutation” in the revised manuscript.

Line 404: The authors discuss agents targeting epigenetic modifications. Please update with more recent (disappointing) findings. 

Response: Thanks for the comment. Based on the comment, we have added two references and updates of disappointing findings of epigenetic therapy, with the explanation of potential challenges of the therapy.

Line 422: Robotic surgery: Please clarify what they mean?

Response: Thanks for the comment. Robotic surgery, also called robot-assisted surgery, allows doctors to perform many types of complex procedures with more precision, flexibility and control than is possible with conventional techniques. Please see at https://www.mayoclinic.org/tests-procedures/robotic-surgery/about/pac-20394974. For breast cancer, robotic surgery offers new surgery and reconstruction option. For more information, please see the website (https://cancer.osu.edu/news/robotic-surgery-approach-offers-new-breast-cancer-surgery-option) and the systematic review paper (https://www.ncbi.nlm.nih.gov/pmc/articles/PMC7015621/, PMID: 32095394). The U.S. FDA first authorized the use of the da Vinci robotic system for specific abdominal surgical procedures in 2000 and then later for radical prostatectomy pelvic surgery in 2001. Since its inception, it is now used in a wide range of urological, gynecological, and general surgical procedures with FDA approval for prostatectomy, hysterectomy, and cholecystectomy. However, the use of robotic surgery in breast cancer is still a new field. We think “robotic surgery” is a well-accepted term and the readers understand what it means. If not, they can search it by themselves easily. In addition, robotic surgery is not discussed as the new uses of technology in the manuscript. Thus, we did not make changes on this.

Part 2. New uses of technology... Why do they select these 4 specific topics and not for example anti-body drug conjugates or artificial intelligence?? Why to discuss mass spectrometry while there is no clinical application.

Response: Thanks for the comment. As we have mentioned in “response to simple summary” as well as in the manuscript, regarding new uses of technology section, well-known technologies such as next-generation sequencing (discussed in the “gene mutation in HR-positive breast cancer” section, and PMID: 23989017, PMID: 23879964, PMID: 30689231), the CRISPR-Cas9 system (PMID: 35879772, PMID: 33031826, PMID: 33926574), artificial intelligence (PMID: 35884503, PMID: 35771379, PMID: 35623118), CAR T-cell therapy (PMID: 35414783, PMID: 32795486, PMID: 33442419), and robotic surgery (PMID: 35815331, PMID: 32095394) have been well and extensively described in many other review papers. Thus, we focused on other four relatively new uses of technology (TIMIST, mass spectrometry, nanoparticle delivery, and PROTAC) in breast cancer in our manuscript.  

As we have discussed in the manuscript, mass spectrometry (MS) remains a reliable approach in the detection of cancer biomarkers and metabolites which are distinguished from normal tissues. It can be used in both clinical and research settings. MS-based proteomics can be used to identify and measure certain proteins, such as INPP4B, CDK1, and ERBB2, which are associated with breast cancer ER status. Thus, the information obtained through MS is useful for the identification of tumor subtypes, as well as for the development of individualized treatment plans.

Line 436: This paragraph is not well written and particular that sentence confusing. Do they mean screening?

Response: Thanks for the comment. The information of tomosynthesis was obtained from the references we cited. It is also mentioned at the U.S. National Cancer institute website: https://www.cancer.gov/types/breast/research#:~:text=Technological%20advances%20in%20imaging%20are,3%2DD%2Dlike%20image. Based on the comment, we have revised the paragraph to be clearer in the revised manuscript.

Line 446 and rest of paragraph: Not of any use for a clinician treating breast cancer. Also, this information is not up-to-date?

Response: Thanks for the comment. As we have mentioned, the readers of the “Cancers” journal not only include clinical physicians, but also include basic scientists and researchers, such as in the fields of breast cancer drug discovery and delivery. As we have discussed in the manuscript, using nanoparticles as a delivery system has been shown to be useful in treating breast cancer, even with metastasis. Thus, we think it is important to discuss it. Based on the comment, we have added three recent references and provided more up-to-date information in the revised manuscript.

Reviewer 2 Report

This manuscript presents a comprehensive review regarding hormone receptor-positive breast cancer. The authors discuss the structures and functions of estrogen and progesterone receptors, the mechanism and resistance of endocrine therapies, somatic gene mutations, epigenetic regulation and abnormally-methylated genes, gene expression-based assays, new targeted drugs, and applications of new technology in detection and treatment. The manuscript is well organized, informative and interesting.

Minor comments:

1. Please add an explanation of “GPER” in the legend of Figure 1.

2. “ER@” in lines 127, 134, 179, 181, and “NF-@B” in line 301: Please state correctly (presumably ERα and NF-κB).

3. Line 177: “raloxifene” is not used for breast cancer treatment, but “toremifene” is used for this purpose.

4. Lines 376, 387−390: “CKD” should be changed to “CDK”.

5. Line 399: “ATK” should be changed to “AKT”.

Author Response

Authors’ Response to Reviewers’ Comments 

We appreciate the Editor and reviewers spending great time to read our manuscript carefully with excellent suggestions. Please see our responses to each comment as follow.

Respond to reviewer #2:

This manuscript presents a comprehensive review regarding hormone receptor-positive breast cancer. The authors discuss the structures and functions of estrogen and progesterone receptors, the mechanism and resistance of endocrine therapies, somatic gene mutations, epigenetic regulation and abnormally-methylated genes, gene expression-based assays, new targeted drugs, and applications of new technology in detection and treatment. The manuscript is well organized, informative and interesting.

Response: We appreciate the excellent comments from the reviewer.

Minor comments:

  1. Please add an explanation of “GPER” in the legend of Figure 1.

Response: Thanks for the excellent comment. The description of GPER was described in lines 126-137 in the original main text. Based on the comment, we have added an explanation of “GPER” in the legend of Figure 1 in the revised manuscript.

  1. “ER@” in lines 127, 134, 179, 181, and “NF-@B” in line 301: Please state correctly (presumably ERα and NF-κB).

Response: We have corrected these errors in the revised manuscript.

  1. Line 177: “raloxifene” is not used for breast cancer treatment, but “toremifene” is used for this purpose.

Response: Thanks for the excellent comment. We have replaced “raloxifene” with “toremifene” and revised the content accordingly in the revised manuscript.

  1. Lines 376, 387−390: “CKD” should be changed to “CDK”.

Response: We have corrected these errors in the revised manuscript.

  1. Line 399: “ATK” should be changed to “AKT”.

Response: We have corrected the error in the revised manuscript.

Round 2

Reviewer 1 Report

The authors did reply to several of my remarks

I don't agree with several of the remarks.

One example: I don't agree they classify breast cancer based on the location of the tumor within the breast. A lobular breast cancer is not named 'lobular' or non-special type based on its localisation but the distinction is based for example on its microscopic growth pattern also using immunohistochemical tests like e-cadherine (and others). The authors show an educational slide which is wrong namely the way they explain this classification.

I am still not convinced why they refer to robot breast surgery in the context of this article, also the story of epigenetics is more complex as what is summarized

Author Response

We appreciate the Editor and reviewers spending great time to read our manuscript carefully with excellent suggestions. Please see our responses to each comment as follow.

Respond to reviewer #1:

The authors did reply to several of my remarks

Response: Thanks for the comment. In our first revision, we have carefully considered the reviewers’ comments and suggestions, and addressed each of the concerns point by point.

I don't agree with several of the remarks.

Response: We respect the reviewer’s comment.

One example: I don't agree they classify breast cancer based on the location of the tumor within the breast. A lobular breast cancer is not named 'lobular' or non-special type based on its localisation but the distinction is based for example on its microscopic growth pattern also using immunohistochemical tests like e-cadherine (and others). The authors show an educational slide which is wrong namely the way they explain this classification.

Response: Thanks for the comment. Based on the comment, we have deleted the sentence “…or anatomical location such as ductal or lobular cancer which can be further characterized by pathological findings” and revised the content accordingly in the revised manuscript (lines 42-45).

I am still not convinced why they refer to robot breast surgery in the context of this article, also the story of epigenetics is more complex as what is summarized.

Response: Thanks for the comment. Based on the comment, we have deleted the words “…and robotic surgery” in the revised manuscript (lines 431-432).

          We agree with the reviewer that “epigenetics is more complex as what is summarized”. In this review, we tried to discuss the complex topic in a better way that is easy to be understood by general readers of the journal, including both health care professionals and research scientists. We have made sure that the content of this topic is correct and appropriately discussed in this manuscript. As we have mentioned in our previous response, to our knowledge, there is no such a review or research paper specifically listed and summarized the commonly mutated and abnormally methylated genes in HR-positive breast cancer, thus, we think this topic is a strength of our review paper.